# Healthcare Workers: Heroes or Victims? Context of the Western World and Proposals to Prevent Violence

**DOI:** 10.3390/healthcare12070708

**Published:** 2024-03-23

**Authors:** Gianpietro Volonnino, Federica Spadazzi, Lina De Paola, Mauro Arcangeli, Natascha Pascale, Paola Frati, Raffaele La Russa

**Affiliations:** 1Department of Anatomical, Histological, Forensic and Orthopedic Sciences, Sapienza University, 00161 Rome, Italy; federica.spadazzi@uniroma1.it (F.S.); lina.depaola@uniroma1.it (L.D.P.); paola.frati@uniroma1.it (P.F.); 2Department of Life, Health and Environmental Sciences, University of L’Aquila, 67100 L’Aquila, Italy; mauro.arcangeli@univaq.it (M.A.); raffaele.larussa@univaq.it (R.L.R.); 3Department of Forensic Medicine, Hospital ‘San Carlo’, 85100 Potenza, Italy; natascha.pascale@ospedalesancarlo.it

**Keywords:** healthcare workers, violence, workplace violence, healthcare systems, prevention

## Abstract

Episodes of direct violence against healthcare workers and social workers represent a worrying and widespread phenomenon in Western countries. These violent attacks, whether verbal or physical, occur in various work environments, targeting professionals working in private facilities, medical practices, or those employed within the National Health System facilities. We conducted a search using a single search engine (PubMed) using the terms “violence against healthcare workers AND Western” for the period 2003–2023, identifying 45 results to which we added to the literature through hand searching. Our review thus analyzed the sector literature to highlight the phenomenon of violence against healthcare workers, particularly in Western countries. We began with an analysis of the problem and then focused on the true purpose of the study, which is to propose new solutions to protect healthcare workers in all work settings. Consequently, we aim to improve both the working environment for healthcare professionals and to enhance the overall healthcare and public health outcomes.

## 1. Introduction

The National Institute of Occupational Safety and Health identifies workplace violence as “any physical assault or attempted assault, threatening behavior, or verbal abuse that occurs in the workplace” [1]. The World Health Organization (WHO) and the International Labour Organization (ILO) have jointly agreed upon a definition of workplace violence: “Incidents in which workers are abused, threatened, or assaulted in circumstances related to their work, including commuting, and which involve an implicit or explicit risk to their safety, well-being, or health” [2]. Thus, episodes of direct violence against healthcare and social care workers can be both physical and verbal. Physical violence involves the use of force resulting in personal injury, while verbal violence entails threats of physical force resulting in physical, mental, spiritual, moral, and social harm. Both forms of violence against healthcare workers represent a concerning and widespread phenomenon in Western countries. Data shows that such events mainly occur in the Emergency Departments (ED) of hospital facilities, primarily targeting emergency healthcare workers. The WHO has estimated that between 8% and 38% of healthcare workers experience physical violence at some point in their careers, and many more are subjected to threats or verbal aggression. Most violence is perpetrated by patients and visitors [3]. However, the literature likely provides data not entirely reflective of reality, with significantly lower values; this seems to stem from the fact that cases of violence often go unreported by victims, resulting in a much lower reporting rate compared to the actual manifestation of the problem [4]. Over the last century, following the world wars and the horror of denied rights, especially the right to health, the Geneva Conventions were ratified. These conventions aim to protect the health of healthcare workers, particularly in times of armed conflict. The Geneva Conventions consist of a series of international law treaties that establish the rights of war victims. The conventions affirm that the right to health and life, and thus physical and mental integrity must never be denied to those caught in armed conflict. The rules of the First and Second Geneva Conventions of 1949 and those established in the Additional Protocols of 1977 and 2005 commit parties to armed conflict to protect healthcare personnel, enabling them to care for the injured regardless of their religion, ethnicity, political affiliation, or affiliation with either side of the conflict. Today, the four Geneva Conventions are universally ratified and apply to all states and parties to conflict. Violation of this agreement constitutes a crime recognized internationally [5]. Despite international law, in recent decades, healthcare workers have too often faced violent attacks not occurring during armed conflicts but predominantly in the workplace and even more so due to the nature of their work. While the value of the work of physicians and all healthcare professionals is legally protected internationally to operate in the best possible way and to protect public health, in everyday news, episodes of direct aggression toward healthcare professionals are increasingly present with incredibly inadequate protections.

## 2. Materials and Methods

The research aims to highlight the working situation of healthcare workers who are increasingly subjected to violent attacks, especially in recent decades. We focused on the literature analysis from 2003 to 2023 using a single search engine (PubMed) and searched for the keywords “violence against healthcare workers” AND “western”. We found 45 results. After excluding documents not relevant to the study’s purpose, the authors analyzed other relevant documents to expand their knowledge of the subject through hand searching. We focused on analyzing this phenomenon only for Western countries and, particularly, on the Italian scenario, which we are more familiar with. In total, 27 documents were included in our research. Figure 1 shows a flow chart of the selection process for these studies. The aim of the study is to analyze the problem of direct violence against healthcare professionals and to seek solutions to propose and implement to protect these healthcare sector professionals. Such proposals would not only improve the work environment but also lead to a significant overall improvement in healthcare and public health outcomes.

## 3. Results

Direct assaults against healthcare workers appear to be a pervasive phenomenon, occurring more frequently as both physical and verbal attacks. Most violent assaults occur within nursing homes and hospitals, particularly in emergency departments, with physicians and nurses being the most affected, especially women [6]. Data on assaults against healthcare workers serving in Emergency Departments are concerning. A study conducted by Gacki-Smith et al. highlighted that in the United States, 25% of their sample of emergency department nurses reported being victims of aggression more than 20 times in the previous three years [7]. A recurring pattern has been observed: the closer the healthcare worker is to the patient or their family members, the greater the likelihood is of encountering aggression. The risk is tripled in these specific cases compared to other healthcare professionals involved in patient care [8]. A study by Aljohani et al. further showed that in more than 70% of cases, violence manifests itself as verbal abuse, while physical aggression occurs in 20% of cases. In more than half of the cases, it is the patient’s relatives who are the perpetrators of the violent event against healthcare workers, with only a quarter of cases involving the patient themselves lashing out at the caregiver [9].

Hesketh et al. [10] identified various types of violence perpetrated against healthcare workers in their study, including attempted assault, physical assault, verbal aggression (insults, humiliation, coercion), verbal sexual harassment, and physical sexual harassment [11]; however, it is noteworthy that the studies analyzed show a significantly lower reporting rate of assaults than the actual occurrence. The primary reason for this appears to be the fear of repercussions from patients and, ultimately, the reluctance to appear weak in the eyes of colleagues. Additionally, the relatively low reporting rate is also attributed to the fact that victims of violence often do not file multiple complaints if assaulted repeatedly but report the event only once. In recent years, there has been increased attention on the impact of assaults, whether reported or not, on the lives and work of healthcare workers. Some studies have demonstrated a correlation between healthcare worker stress, including diagnoses of burnout, and negative outcomes for the patients under their care [12]. Violence perpetrated against healthcare workers can lead to dissatisfaction and work-related stress, ultimately contributing to burnout [13]. To clarify, healthcare workers subjected to any form of aggression may experience job dissatisfaction and work-related stress, which ultimately affects their ability to operate and treat patients effectively. This creates an unhealthy cycle (Figure 2) in which user violence against healthcare workers inevitably rebounds to users in the form of deteriorating care standards [14].

### 3.1. In the World

Workplace safety and the increasing number of assaults on healthcare workers constitute a global resonant issue. Violence against healthcare workers is a widely spread phenomenon globally and unfortunately seems to be occurring with growing numbers. The scale of this phenomenon is evidenced by a greater number of scientific documents in the literature on the subject compared to previous decades. Most of these studies particularly concern the category of nurses, and they are considered as the most vulnerable category according to the previously mentioned equation of greater proximity to the patient equals greater risk of aggression. Nurses, in fact, represent the frontline in healthcare action, maintaining professionalism during situations such as having to deal with patients in pain, patients who are stressed, and often difficult to manage, which can frequently lead to tense situations [15,16]. We have observed, however, that episodes of violence directed towards healthcare personnel are more prevalent in Asian countries compared to Europe. This could be due to cultural differences, working contexts, significantly lower numbers of healthcare workers compared to demand, and higher workloads for doctors and nurses [17]. The World Health Organization has also drawn attention to this issue, which not only affects workplace safety and consequently the quality of care for all those seeking healthcare providers, but also constitutes a work-related risk that should not be underestimated and should be countered with appropriate and specific preventive measures. It has been observed that the exposure to violent events for healthcare professionals is more than ten times higher compared to other sectors [18]. A systematic review conducted in 2014 by Spector et al. analyzed data from 160 healthcare facilities worldwide, involving over 150,000 nurses. This study, among others, sheds light on the issue of assaults and highlights how geographically there are areas that experience a higher frequency of violent incidents. For example, in Anglo-Saxon countries, there is a greater culture of reporting or a lesser acceptance of the phenomenon, both generally and among professionals. In contrast, there are other parts of the world characterized by a lower frequency of violence against healthcare workers or where there may be lower reporting or greater acceptance, such as in Europe [19].

### 3.2. In USA

In the United States, violent attacks against healthcare workers and healthcare facilities manifest themselves primarily as incidents of armed aggression and fierce shootings, often resulting in injuries to other individuals. According to the U.S. Bureau of Labor Statistics, workplace assaults from 2011 to 2013 ranged from 23,540 to 25,630 annually, with over 70 percent occurring in healthcare and social service settings (OSHA) [20]. The emergency department, mental health and long-term care providers are among the most frequent victims of patient and visitor attacks. Many studies conducted across the ocean in emergency and urgent care departments reveal that more than two-thirds of healthcare workers experienced at least one violent aggression in the workplace in the last year. The American Medical Association (AMA) has evaluated that healthcare workers are up to four times more likely than other private sector workers to suffer workplace injuries resulting in days of illness and consequent absence from work, inevitably impacting public health [21]. To address this issue, the Health Alliance for Violence Intervention (HAVI) was established in 2018 to promote collaboration between hospital facilities and healthcare professionals to prevent acts of violence against healthcare workers and disseminate nationwide awareness and prevention programs in hospital settings [22]. Additionally, in some U.S. states such as California and Pennsylvania, there have been considerations to increase sanctions, with a deterrent intent, for direct assaults against healthcare workers [23]. 

### 3.3. In Italy

In Italy, during the triennium from 2019 to 2021, more than 4800 cases of violence, assaults, and threats against healthcare and social healthcare personnel were identified by INAIL (National Institute for Insurance against Accidents at Work), with an average of about 1600 cases per year [24]. Even in Italy, as in other Western countries, patients and their relatives are the ones most commonly responsible for attacking healthcare workers. INAIL reports that approximately 10% of injuries reported by healthcare workers, especially those working in hospital facilities, are attributable to assaults. To raise awareness about this issue, Italy established a National Day of Education and Prevention against Violence towards Healthcare and Social Healthcare Workers on March 12th of each year through a decree by the Minister of Health dated 27 January 2022 [25,26]. Recently, there has been an increasing number of cases of aggression against healthcare professionals of all specialties and working in various settings, both public and private. Here are some examples: 29 November 2022—Southern Italy: A patient fatally shot a cardiologist because the doctor refused to renew the patient’s driving license, despite known cardiac problems that made him unfit.5 October 2023—Central Italy: An immunologist was assaulted and violently beaten in his medical office in Rome by a patient who believed the doctor had prescribed the wrong therapy to treat a spinal infection. The doctor was admitted with severe craniofacial trauma, fracture of the left orbital floor, and nasal septum rupture.23 October 2023—Northern Italy: A general practitioner was assaulted and injured during office hours by a patient to whom he had denied an unnecessary prescription.

The INAIL data are concerning; they indicate that more than 60% of assaults, or at least those reported, occur in the northern regions of Italy, with minimal percentage differences. Additionally, the reported cases allow us to extrapolate another data point on the types of injuries healthcare workers sustain; injuries are mainly contusions and sprains, particularly localized in the head and upper limbs, with wounds or fractures in 16% of cases. Additionally, the location of the injuries reflects the severity of the attacks. Striking someone in the head implies not just an attempt to intimidate but also a clear and evident intent to cause harm. Most of these events have also been observed to take place within hospital facilities and care facilities scattered throughout the territory. Further evaluation of the type of aggression sadly points to the issue of gender-based violence. Unfortunately, it is clear that there is a growing trend of aggression, especially against female healthcare workers. Women are indeed the population of healthcare workers most affected, accounting for up to 71% of total assaults. Most assaults target workers under the age of 50 years, and the most affected profession, representing more than a third of cases, is that of healthcare technicians, including nurses and professional educators engaged in educational and rehabilitation services. Direct assaults against medical personnel represent around 3% of cases; these data naturally do not include generic healthcare workers and freelancers who are not accounted for by INAIL. Lastly, in Italy, as in the rest of the Western world, violence is committed directly by patients in 49% of cases who are directly involved in medical treatments and thus in direct contact with healthcare workers, and in the remaining 51%, it is committed by their relatives [27].

### 3.4. Aggressions during the SARS-CoV-2 Pandemic

During the SARS-CoV-2 pandemic, as highlighted in the study by Banga et al., there was an increase in aggressions against healthcare workers in 2022 compared to previous years, especially compared to 2020 when access to healthcare facilities was limited due to the pandemic. This increase can be related to several factors [11]. It was noticed that with the decrease in direct contact between healthcare workers and patients or their relatives, the number of aggressions also significantly decreased. However, it is important to note that such data always exclude freelance medical professionals or uninsured nurses [28].

Another important factor is represented by the esteem and hope placed in the medical sector and the scientific expertise held by healthcare workers during a historical moment that no one could have imagined having to live through. During the COVID-19 pandemic, doctors were indeed hailed as “heroes”; they were celebrated for their work, for the daily risks they took to save the lives of infected patients and were applauded from balconies during lockdown periods. The esteem that the entire population had for healthcare workers was at an all-time high; unfortunately, after the emergency phase passed, people began to forget about their heroes, and these heroes were often relegated to victims, and too often to “sacrificial” victims. The same voices that once celebrated the tireless efforts of healthcare workers quickly turned into verbal aggressions, insults, and offenses; the same hands that once applauded became weapons to be used against healthcare workers. These data reflect well the influence that a pandemic like the one experienced in recent years can have and has had on the human psyche [29]. Initially, during the emergency phase of the pandemic, doctors and healthcare workers were widely celebrated as heroes for their commitment to managing the crisis and saving lives. However, as the acute phase of the emergency passed, this esteem and gratitude began to diminish, transforming into verbal aggressions, insults, and physical attacks against healthcare personnel. This change can be attributed to public frustration, fear of contagion from healthcare workers, dissatisfaction with imposed restrictions, and ignorance about the severity of the pandemic, as emphasized by Journal of Nursing & Health [30]. This shift in attitude had a significant impact on the mental health of healthcare workers. They found themselves facing not only the threat of the virus but also the lack of adequate individual protection, increased workload, and hostility from the population. This contributed to greater emotional instability, manifested through symptoms of depression and anxiety [31]. The lack of patient trust in healthcare personnel further exacerbated the situation, leading to a reduction in healthcare performance and quality of care [32]. All of this created a stressful and insecure work environment for healthcare workers, contributing to a decrease in their ability to provide high-quality care and an increase in psychological problems among healthcare personnel, such as depression and anxiety [33]. In summary, the COVID-19 pandemic has exacerbated the risks of aggression and violence against healthcare workers, highlighting the need for greater protection and support for those working in the healthcare sector during health emergencies [34].

### 3.5. Legislative Measures

To address this issue, which has affected the entire planet, institutions have decided to disseminate recommendations and enact legislative measures to protect healthcare workers and make them aware of the general situation and how to act in case of direct violence against them. The European Agency for Safety and Health at Work (EU-OSHA) is particularly involved in promoting campaigns aimed at raising awareness among the European population regarding health and safety in the workplace [35]. Attention to the issue is high. In Italy, it was published in the Official Gazette n. 224, in Law 113/2020 that “Provisions on safety for those exercising healthcare and socio-health professions in the performance of their functions”, with the aim of protecting workers in the healthcare sector [26]. This law also supports the establishment of a national observatory on the safety of those practicing health and social-health professions (ONSEPS) with the Decree of 13 January 2022 [36]. ONSEPS is tasked with observing episodes of violence and implementing measures to raise awareness, prevent violence, and protect victims. In addition, the law intervenes primarily in the area of providing more information to citizens and healthcare professionals about the issue at hand to reduce risk factors in the most exposed environments, disseminate good behavioral practices, improve communication with patients; secondly, it addresses the issue of harsher penalties, intervening on Article 583-quarter of the Criminal Code, for those who cause serious or very serious personal injuries to anyone carrying out care activities in the exercise of their function, identifying the fact as aggravating; it makes offenses of assault (Article 581 of the Italian Criminal Code) and personal injury (Article 582 of the Italian Criminal Code) against healthcare workers prosecutable ex officio; it also provides security measures aimed at increasing surveillance in healthcare environments and improving safety plans; and finally, it provides for an administrative sanction of an amount ranging from 500 to 5000 euros for anyone who engages in violent, insulting, offensive, or harassing conduct against healthcare or socio-healthcare professionals as well as against anyone providing care, healthcare assistance, or rescue activities in public or private healthcare and socio-healthcare facilities. Additionally, on 30 March 2023, a Legislative Decree was issued that, in Article 16, deals with provisions relating to combating acts of violence against healthcare personnel and amends the Article 583-quarter of the Criminal Code. These amendments further increase penalties for violence committed against healthcare workers or anyone performing auxiliary activities. The law converting Decree-Law 34/2023 strengthens the fight against personal injuries to healthcare workers by providing for imprisonment ranging from two to five years. Personal injury to a doctor/healthcare worker is therefore considered an aggravating circumstance of the offense of personal injury. If the personal injuries to the professional are “serious” or “very serious”, the penalty can increase to up to 10 years and 16 years of imprisonment, respectively [37]. 

## 4. Discussion

The data presented and the literature in the field highlight a pervasive phenomenon, with an increasing incidence at both national and international levels. The spread of this problem makes it clear and mandatory to implement preventive measures against direct violence towards healthcare workers and to raise awareness among the population. Furthermore, more impactful support measures are needed for healthcare workers who are victims of violence, including in actuarial settings. Direct violence against healthcare workers appears to be indicative of underlying issues, and we have sought to better understand what drives a patient or a family member to assault a healthcare worker. This question seems to be at the core of any discussion on the topic. The underlying issue of this phenomenon also seems to be the devaluation of the role of the physician, seen as a mere laborer, fulfilling more or less appropriate requests from a medical and legal standpoint, a servant, and a receptacle for frustrations about the patient’s or their family’s health status. From studying the literature in the field, we have noticed, without significant differences between Western countries, that one of the reasons behind violent behavior is resentment or revenge for a perceived wrong (whether real or not) and the perception of inadequate patient care by healthcare workers. Therefore, the first point to implement and improve is the dialogue between healthcare workers and patients or their caregivers. A better explanation of the clinical situation and sharing of treatment timelines and choices, to the extent possible, could certainly help the understanding of patients waiting for care and could make the work of healthcare workers easier. For example, it would be useful to integrate liaison figures between patients and healthcare workers into emergency departments, who can provide psychological and logistical support to those seeking care and their caregivers, as well as alert healthcare workers to situations where anger and fear about the perceived dangerous and inadequately addressed situation could escalate into episodes of violence. Improving the quality of doctor/nurse-patient communication is also necessary, ensuring that the patient has a better understanding of their health status and what they will face in terms of diagnostic tests and treatment options. Improved communication would allow for the establishment of the trust relationship that has been deteriorating over time due to speculation about the professional responsibility of the physician, which especially in Italy, continues to be perpetuated by civil litigation lawyers with unfounded disputes. This communication between the creditor and the debtor of the healthcare service would lead to a consequent improvement in adherence to treatment, with positive results and a restoration of the value of the professional figure. We have also found that some hospital facilities have implemented precautionary measures consisting of an increase in surveillance systems. There has been an increase in the number of security personnel and expanded coordination with law enforcement. Alarm and video surveillance systems have been installed to record 24 h a day, and thus also serve as a deterrent for aggressive actions. “Mass” access to hospital premises has been limited to allow security personnel more adequate control and to avoid confusion in the areas designated for patient primary care. It would therefore be appropriate to improve the triage phase in patient reception, avoiding long waits, mostly in austere, crowded, depersonalized environments without the possibility of communication with healthcare staff but only through electronic means, which is not always immediately understandable especially by the elderly or those with lower socio-cultural levels. Healthcare facilities should be encouraged to adopt policies to facilitate reporting, with reports from healthcare workers readily available and fillable. It would be useful to promote the implementation of training courses for healthcare workers to implement appropriate forms of communication in emergency situations and to prevent and respond to threats or violent assaults in the workplace. It would also be desirable to avoid “face-to-face” healthcare–patient situations, especially in outpatient settings, private medical practices, isolated facilities without internal surveillance, so that the healthcare worker who finds themselves in a dangerous situation is never alone and without psychological and physical support. Precisely because of this, in recent years, the idea of “gathering” medical practices, especially those of territorial doctors, in so-called “family homes” or multi-specialist outpatient clinics with scheduled and limited user access, is advancing; this would allow for the presence of multiple healthcare workers in the same medical office with associated reference figures (medical secretaries and security personnel) who could, if not prevent, at least better manage unpleasant situations that could escalate into violent assaults. From an actuarial standpoint, the idea that insurance measures aimed at compensating for physical or psychological harm, closely related to the issue under study, are also necessary is gaining ground. These insurance measures should guarantee at least the right to compensation for harm to individuals against healthcare workers, who too often see their complaints dismissed and are therefore not inclined to report further. It would therefore seem desirable to have a much broader, standardized action plan implemented throughout the national, and why not, European territory, aimed at providing greater protection to this category of professionals, who not long ago were considered “heroes”, but are increasingly being relegated to victims today.

## 5. Conclusions

Violence against healthcare workers has become a widespread phenomenon in both public and private work settings in recent years. This study aimed to identify common critical aspects across different countries and propose preventive and compensatory measures.

The underlying causes of this phenomenon appear to be various issues: the devaluation of the healthcare worker’s role, patients’ and families’ fear of illness, lack of adequate communication between doctor and patient, and shortage of staff in emergency departments.

Therefore, the ideal approach would be to enhance communicative aspects both qualitatively, through specialized training for healthcare workers, and quantitatively, by integrating psychological support personnel into healthcare environments. This would serve the dual function of identifying and preventing risky situations and increasing patient trust in the attending physician, thereby alleviating psychological stress such as long waiting times.

It is also proposed to increase alarm systems and video surveillance in public facilities and to transform private medical offices into associated structures, ensuring that healthcare workers are never alone in dealing with critical situations.

Finally, we propose compensatory measures in the form of insurance coverage for healthcare workers in case of physical or psychological harm due to aggression, with the aim of increasing reporting rates among assaulted healthcare workers.

## 6. Limitations

Our study has limitations that we want to bring to the reader’s attention to make our work clearer and more usable. Violence against healthcare workers is a widespread phenomenon, but as mentioned earlier, reports are lacking for several already analyzed reasons. Therefore, the first limitation lies in the data. They are inherently underestimated from the outset and should be considered as such. Another limitation is represented by our research conducted only in a single search engine, which, however, provided us with data from all parts of the world, so it was not inherently exclusive. Another limitation is represented by our choice to analyze only data from countries in the so-called Western world, where this phenomenon has been more prevalent in recent years. Among Western countries, we then decided to focus on the New and Old World; therefore, we highlighted the phenomenon in the USA and then in Europe, where Italy was our most logical choice due to affiliation and familiarity. Despite these limitations, we hope that our document can provide an impetus for further research and can serve especially to identify preventive measures and appropriate corrective interventions commensurate with the phenomenon of violence against healthcare workers.

## Figures and Tables

**Figure 1 healthcare-12-00708-f001:**
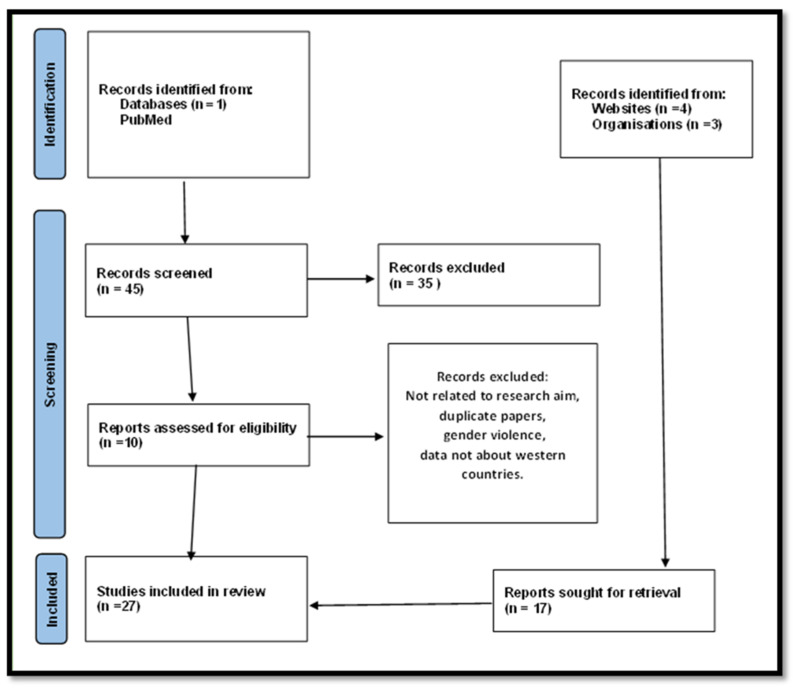
PRISMA flow chart.

**Figure 2 healthcare-12-00708-f002:**
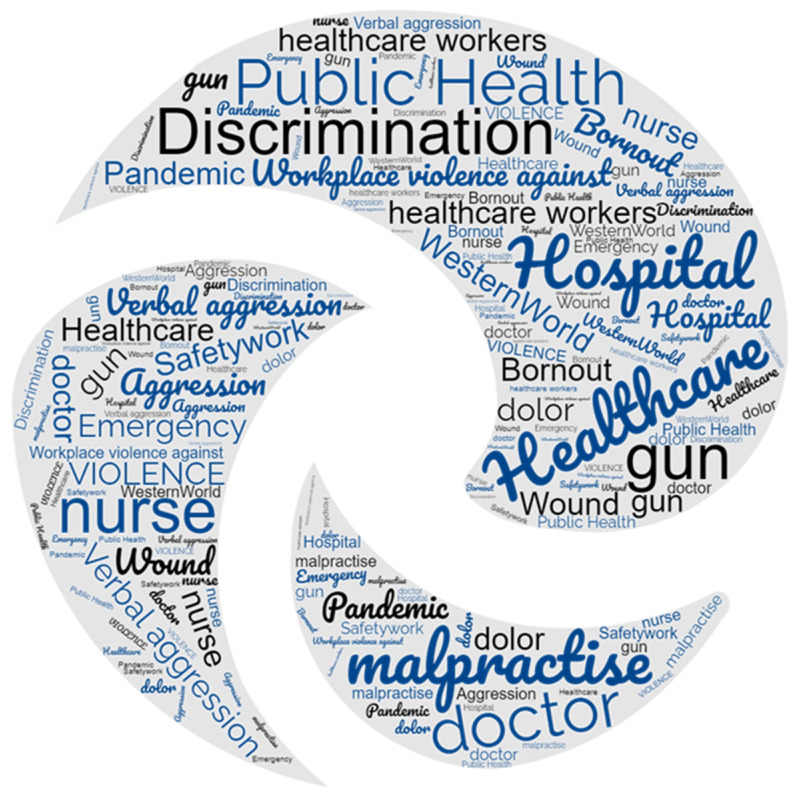
Vicious cycle of violence.

## Data Availability

The datasets used and/or analysed during the current study are available from the corresponding author upon reasonable request.

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
