# Peer review of "Healthcare Workers: Heroes or Victims? Context of the Western World and Proposals to Prevent Violence"

_healthcare, 2024, doi:10.3390/healthcare12070708_

Round 1

Reviewer 1 Report

Comments and Suggestions for Authors

The paper deals with the issue of violence against healthcare workers and social workers. The paper offers the analysis of literature published from 2003 to 2023. Using a single search engine (PubMed), 45 documents were found, among which 27 documents were the subject of analysis. According to the conducted research, most violent assaults described in the literature occur within nursing homes and hospitals, particularly in emergency departments, with physicians and nurses being the most affected, especially women” (p. 3). Moreover, the majority of existing studies “particularly concern the category of nurses” (p. 4). The paper offers a systematization of the knowledge that can currently be found in the literature and is therefore of some value, but it does not bring new knowledge about violence against healthcare workers and social workers. The section about aggressions during the SARS-CoV-2 pandemic is of particular interest. The new Italian legislation addressing the issue of violence directed against healthcare is also accurately analysed  (p. 7).

Comments on the Quality of English Language

The quality of English language is sound.

Author Response

Reviewer: The paper deals with the issue of violence against healthcare workers and social workers. The paper offers the analysis of literature published from 2003 to 2023. Using a single search engine (PubMed), 45 documents were found, among which 27 documents were the subject of analysis. According to the conducted research, most violent assaults described in the literature “occur within nursing homes and hospitals, particularly in emergency departments, with physicians and nurses being the most affected, especially women” (p. 3). Moreover, the majority of existing studies “particularly concern the category of nurses” (p. 4). The paper offers a systematization of the knowledge that can currently be found in the literature and is therefore of some value, but it does not bring new knowledge about violence against healthcare workers and social workers. The section about aggressions during the SARS-CoV-2 pandemic is of particular interest. The new Italian legislation addressing the issue of violence directed against healthcare is also accurately analysed  (p. 7).

Author: Thank you for your review and advices.

Reviewer 2 Report

Comments and Suggestions for Authors

Dear colleagues!

Thank you for the interesting and, dare I say it, poignant topic of research. Indeed, issues of sacrifice of medical activity have always been a subject of discussion and compromise solutions.

I have a few questions

1. What were the limitations of the study?

2. Why are your countries so geographically isolated: the USA and Italy? Did your study include participants from Asian and African countries?

3. What recommendations would you make based on the results of the study for the health care system and legal regulations?

Author Response

Dear Reviewer.

Thank you for the review. 

We clarified your questions in the "Limitations" section. Unfortunately, such data are very underestimated and we have focused on a reference country (USA) and our native to collect the most accurate data.

Point-by-point response to reviewers and editors 

Dear Reviewer, thank for your work. Here is our point by point response.

Comment: 1. What were the limitations of the study?

Response: We added “Limitations” section

Comment: 2. Why are your countries so geographically isolated: the USA and Italy? Did your study include participants from Asian and African countries?

Response: As explained in “Limitations” section, we analyzed mostly Usa and Italy, in order to get a parameter in Western Countries

Comment: 3. What recommendations would you make based on the results of the study for the health care system and legal regulations?

Response: The continuous forms of aggression against health workers have sensitized legislators in Italy, with interventions that provide specific penalties for injuries caused to health workers

Reviewer 3 Report

Comments and Suggestions for Authors

The study aims to examine the working conditions of healthcare workers, who have experienced a rising trend of violent attacks, particularly in recent decades. The study included  literature analysis using PubMed, uncovering 27 relevant studies. Despite the intriguing subject matter, the article exhibits several shortcomings, particularly in the areas of introduction, methodology, and, notably, the discussion. Major revision of the article is needed. My major comments on this topic are listed as follows;

1.      The introduction should be revised to minimize redundancy in the description of violence. This section should convey more info from other countries rather than focusing  specific countries.

2.      There is no specific research objectives in introduction . Research questions and the study's significance should be elaborated.

3.      The rationale for limiting search terms, specifically "violence against healthcare workers" AND "western." should be provided. The authors need to justify why these terms were chosen. Additionally, a rationale for the broad publication date range from 2003 to 2023 is needed .

4.      Why did the authors prefer to use only PubMed as the search engine. AS It is not an inclusive database especially on this topic, it may limit the review’s effect. This issue should be clarified and discussed.

5.      The inclusion and exclusion criteria for including studies have not been detailed. This should be added.

6.      The method section should outline criteria for assessing the quality of included studies.

7.      The discussion lacks citations to previous studies and should be rewritten from scratch. It seems like it only based on the authors' ideas. The discussion needs to be more thorough, coherent, and well-supported.

Author Response

Dear Reviewer,

Thanks for the review and valuable advice. We have clarified the greater concentration of this phenomenon on the Italian scene, especially in light of the proposed new laws

Point-by-point response to reviewers and editors 

Dear Reviewer, thank for your work. Here is our point by point response.

  1. The introduction should be revised to minimize redundancy in the description of violence. This section should convey more info from other countries rather than focusing specific countries.
  2. There is no specific research objectives in introduction . Research questions and the study's significance should be elaborated.
  3. The rationale for limiting search terms, specifically "violence against healthcare workers" AND "western." should be provided. The authors need to justify why these terms were chosen. Additionally, a rationale for the broad publication date range from 2003 to 2023 is needed .

Response: 1, 2, 3 -We added “Limitations” section, specifying the greater concentration on the Italian scenario both in terms of practicality and because a legislative reform is also being implemented aimed at changing the penalties in cases of aggression against health personnel

  1. Why did the authors prefer to use only PubMed as the search engine. AS It is not an inclusive database especially on this topic, it may limit the review’s effect. This issue should be clarified and discussed.
  2. The inclusion and exclusion criteria for including studies have not been detailed. This should be added.

Response: 4, 5- The choice of only one search engine is due to the fact that it allowed us to receive results from multiple geographical sources

  1. The method section should outline criteria for assessing the quality of included studies.
  2. The discussion lacks citations to previous studies and should be rewritten from scratch. It seems like it only based on the authors' ideas. The discussion needs to be more thorough, coherent, and well-supported.

Response: 6, 7 – We added “limitations” and “conclusions” sections

Round 2

Reviewer 3 Report

Comments and Suggestions for Authors

Authors responded my comments adequately.